# Role of MicroRNAs in Neuroendocrine Prostate Cancer

**DOI:** 10.3390/ncrna8020025

**Published:** 2022-03-30

**Authors:** Amritha Sreekumar, Sharanjot Saini

**Affiliations:** Department of Biochemistry and Molecular Biology, Augusta University, Augusta, GA 30912, USA; amsreekumar@augusta.edu

**Keywords:** microRNAs, neuroendocrine differentiation, castration-resistant prostate cancer

## Abstract

Therapy-induced neuroendocrine prostate cancer (t-NEPC/NEPC) is an aggressive variant of prostate cancer (PCa) that frequently emerges in castration-resistant prostate cancer (CRPC) under the selective pressure of androgen receptor (AR)-targeted therapies. This variant is extremely aggressive, metastasizes to visceral organs, tissues, and bones despite low serum PSA, and is associated with poor survival rates. It arises via a reversible trans-differentiation process, referred to as ‘neuroendocrine differentiation’ (NED), wherein PCa cells undergo a lineage switch and exhibit neuroendocrine features, characterized by the expression of neuronal markers such as enolase 2 (ENO2), chromogranin A (CHGA), and synaptophysin (SYP). The molecular and cellular mechanisms underlying NED in PCa are complex and not clearly understood, which contributes to a lack of effective molecular biomarkers for diagnosis and therapy of this variant. NEPC is thought to derive from prostate adenocarcinomas by clonal evolution. A characteristic set of genetic alterations, such as dual loss of retinoblastoma (*RB1*) and tumor protein (*TP53*) tumor suppressor genes and amplifications of Aurora kinase A (*AURKA*), *NMYC*, and *EZH2*, has been reported to drive NEPC. Recent evidence suggests that microRNAs (miRNAs) are important epigenetic players in driving NED in advanced PCa. In this review, we highlight the role of miRNAs in NEPC. These studies emphasize the diverse role that miRNAs play as oncogenes and tumor suppressors in driving NEPC. These studies have unveiled the important role of cellular processes such as the EMT and cancer stemness in determining NED in PCa. Furthermore, miRNAs are involved in intercellular communication between tumor cells and stromal cells via extracellular vesicles/exosomes that contribute to lineage switching. Recent studies support the promising potential of miRNAs as novel diagnostic biomarkers and therapeutic targets for NEPC.

## 1. Introduction

According to recent global cancer statistics, prostate cancer (PCa) remains the leading cause of cancer deaths among men in 48 countries [1]. Prostate cancer is the most commonly diagnosed solid organ tumor among U.S. men. The first line of treatment against this disease includes radiation therapy, androgen deprivation therapy (ADT) and surgery. The treatment regimen is decided based on PSA level, grade, and stage of cancer. In 1972, Huggins and Hoggs introduced the concept of androgen dependence in prostate cancer. Since then, ADT has gained prominence in PCa treatment [2]. ADT is used along with, after, or before radiation treatment, especially when there are chances of recurrence. Approximately 37% of PCa patients worldwide receive ADT. A substantial number of men manage the cancer with continuous ADT for more than 6 months during the course of their disease [3]. ADT is found to be initially effective; however, 2–3 years into ADT, the disease evolves to castration-resistant prostate cancer (CRPC) [4]. The resistance to ADT evolves through a variety of mechanisms, including constitutive AR signaling through AR splice variants, ligand-dependent activation of AR, and amplification, overexpression, and mutations in the AR genes [5]. In CRPC, PCa cells are insensitive to ADT, nevertheless androgen receptor (AR) signaling is still active. Highly potent second-generation AR pathway inhibitors (APIs) such as enzalutamide and abiraterone are used as next line of therapy and are effective initially. However, in a significant proportion of patients (~30%), API resistance develops owing to heterologous mechanisms such as restored/bypass AR signaling or evolution to an AR-independent state. Therapy-induced neuroendocrine prostate cancer (t-NEPC/NEPC) arises via a reversible trans-differentiation process, referred to as ‘neuroendocrine differentiation’ (NED) [6,7], wherein PCa cells undergo a lineage switch and exhibit neuroendocrine features, characterized by the expression of neuronal markers such as enolase 2 (ENO2), chromogranin A (CHGA), and synaptophysin (SYP). t-NEPC is also seen in PCa patients after primary docetaxel therapy [8,9]. t-NEPC is an extremely aggressive and invasive variant of prostate cancer that progresses rapidly and metastasizes to visceral organs, tissues, and bones [10], despite low serum PSA [11,12]. The molecular and cellular mechanisms underlying PCa NED are not clearly understood, which contributes to the lack of effective biomarkers for diagnosis and prognosis of this PCa variant. A reliable means for predicting ADT treatment resistance and evolution to NEPC would facilitate individualized patient treatment regimens.

## 2. Therapy Resistance in PCa and Evolution of NEPC: Interplay between Genetics and Epigenetics

NEPC has a unique gene expression profile that is distinguishable from primary PCa and CRPC as it harbors a different set of genetic alterations and epigenetic states [13]. Several groups have reported the genetic and epigenetic changes underlying NEPC [13,14,15,16,17,18]. NEPC is reported to be derived via clonal evolution from adenocarcinomas [13]. A characteristic set of genetic events drive the transition of prostate adenocarcinomas to NEPC. These genetic alterations include frequent *TMPRSS2-ERG* rearrangements [17], dual loss of retinoblastoma (*RB1*) and tumor protein (*TP53*) tumor suppressor genes and amplifications of Aurora kinase A (*AURKA*), *NMYC* and *EZH2* [13,14,15,16,18]. RB1 restricts the expression of E2F1, a transcription factor (TF) that regulates G1–S transition in the cell cycle. The loss of *RB1* releases E2F1 that directly regulates the expression of CHGA, SYP, and NSE, thereby promoting NEPC [19]. AURKA interacts with N-Myc and stabilizes it [15,16,20], while N-Myc co-binds with E2F1 to drive transcription of the neuronal genes *CHGA*, *SYP*, and *ENO2* [19]. The neural transcription factors BRN2 [21] and BRN4 [22] play important roles in this trans-differentiation. A recent study suggests that AR activity is maintained in NEPC as tumors adopt alternative lineage identities [23]. This study showed that upon emergence of NEPC changes in chromatin architecture guide AR transcriptional rerouting and that EZH2 cooperates with AR to transcriptionally modulate stem cell and neuronal gene networks [23]. Interestingly, two recent studies suggest that NEPC is not a single disease state; rather, it can be stratified into distinct subtypes based on the expression of neuronal factors (ASCL1 and NEUROD1) [24] and splicing factors (SRRM3 and SRRM4) [25]. Although these studies have characterized the key genomic and epigenetic alterations driving NED, we are still far from a complete understanding of key molecular alterations. MicroRNAs (miRNAs) constitute important epigenetic players driving NED [26,27] (discussed in the next section). In addition to miRNAs, other non-coding RNAs, such as long non-coding (lncRNA) and piwi-associated RNAs, have also been reported to be involved in NEPC trans-differentiation [27].

## 3. MicroRNAs as Key Epigenetic Players in Neuroendocrine Prostate Cancer

MicroRNAs (miRNAs) are small non-coding RNAs that are important gene regulators, acting primarily via post transcriptional gene silencing. miRNAs have sequence-specific interactions with 3′-untranslated regions (UTRs) of cognate mRNA targets [28]. mRNA harbors multiple target sites for miRNAs and diverse miRNAs could bind to the same target sites. Therefore, miRNAs exercise pleiotropic regulation and are part of a plethora of signaling pathways [29]. miRNAs are involved in several cellular functions such as cell growth, differentiation, and apoptosis [30]. As miRNAs play key roles in major cellular processes, their altered expression leads to diseased states, including cancer. Studies have shown that the correlation of altered miRNA expression and prostate cancer [31]. miRNAs can act as oncogenes (gain-of-function) or tumor suppressor genes (loss-of-function) based on the role they play in tumorigenesis [32,33]. Essentially, oncogenes/oncomiRs are miRNAs that repress the expression of tumor suppressor genes such as *TP53*, *PTEN* and *RB1*, while miRNAs that repress oncogenes such as *MYCN* and *AURKA* are tumor suppressors [31].

miRNA levels can be dysregulated by various mechanisms, such as the methylation of their promoters, by other tumor promoting factors, or by genomic loss or deletion [34]. Evidently, miRNAs are potent mediators between genetic programs and phenotypic manifestations in tumor cells. Moreover, microRNAs mediate the crosstalk between tumor cells and the tumor microenvironment (TME). miRNAs are reported to be transported out of the cell via extracellular vesicles such as exosomes to regulate tumor growth [35]. In this review, we discuss the regulatory role of miRNAs in NEPC.

Recent studies suggest that there may be spectrum of cellular differentiation states as AR+ prostate adenocarcinomas evolve to AR-independent NE states [14,25]. Prostate tumors progress from AR+ adenocarcinomas to AR- therapy-induced NEPC via an extensive network of transcriptional reprogramming events that lead to the expression and maintenance of neuronal features [36]. The molecular networks underlying emergence of these neuronal states are still being deciphered. However, many factors have been associated with the emergence and progression of neuroendocrine differentiation in adenocarcinomas including the downregulation of AR and/or AR-mediated signaling and upregulation of specific oncogenic signaling. In addition, the epithelial-to-mesenchymal transition (EMT)-induced cellular plasticity and stem cell signaling pathways promote the progression to neuroendocrine variant prostate cancer [36]. In discussing the roles of microRNAs in NEPC, we discuss how miRNAs influence various cellular processes that promote neuroendocrine trans-differentiation (Figure 1 and Figure 2). We are also evaluating microRNAs as plausible biomarkers that are important for diagnostics.

### 3.1. MicroRNAs Regulating Oncogenes Implicated in NEPC

The genetic and epigenetic mutational landscapes of NEPC tissues are substantially different from those in the CRPC tissues [13]. NEPC shows a high incidence of the loss of *RB1* and *TP53*, as well as frequent concurrent amplification of *MYCN* and *AURKA*, a mitotic kinase expressed during the G2 phase of cell division [37,38]. Gene amplification of *MYCN* and the upregulation of AURKA are observed in approximately 40% of NEPC. These genes are reported to be regulated by miRNAs. miR-34a, miR-101, and let-7 have been shown to reduce N-MYC expression and decrease the proliferation of *NMYC*-amplified cells [39,40,41]. N-MYC stabilizes AURKA via FBXW7. Li et al. reported that *TP53* mutation upregulates miR-25, which subsequently downregulates an ubiquitin E3 ligase, FBXW7, which degrades AURKA. In the absence of FBXW7, AURKA expression goes unchecked (Figure 1) [42]. miRNAs are known to mediate differential gene expression. N-MYC overexpression is found to deregulate ATM expression based on the androgen dependence of the cells with the help of miR-421. C4-2 cells devoid of AR showed downregulated miR-421 and augmented ATM expression, whereas in androgen-dependent LNCaP cells, upregulated miR-421 repressed ATM expression (Figure 1) [43].

We reported that miR-106a~363 inhibition led to the induction of neuronal genes in CRPC cell lines [26]. Importantly, miR-106a~363 was found to pleiotropically regulate multiple NEPC drivers, including N-Myc, E2F1, STAT3, and AURKA (Figure 1) [26]. The miR-106a~363 family is located on chromosome Xq26.2, a region known to be associated with CRPC-NE-specific copy number alterations [14].

Next-generation miRNA sequencing has identified miR-652 associated with increased tumor growth, migration, and invasion [44]. It directly targets PPP2R3A that encodes “B” subunit of the serine/threonine phosphatase PP2A. PP2A is a serine/threonine phosphatase that regulates multiple pathways associated with prostate cancer. When PP2A activity is inhibited in androgen-independent cells via miR-652, multiple pathways are altered. For example, the inactivated PP2A complex fails to repress AKT/ERK kinases, causing ERK phosphorylation and the subsequent ZEB1 and N-cadherin upregulation that consequently trigger the EMT (Figure 2). Phosphorylated AKT also activates β-catenin. In androgen-deprived cells, activated β-catenin transcribes proteins that induce neuroendocrine differentiation [44].

The 8q24 chromosomal locus is a region of high PCa susceptibility that carries genetic variants associated with high risk of PCa incidence. This region harbors PVT1 gene, which encodes a miRNA cluster. This gene undergoes frequent amplification during cancer [45,46]. One of the miRNAs, miR-1205, is found to be downregulated in NEPC [47]. It directly targets FRYL (Frylike) transcription coactivator. FRYL is speculated to have a role in dendritic branching; hence, it is hypothesized to induce NED. Nevertheless, the studies reported that PCa undergoes FRYL-independent NE differentiation. However, the mechanism by which miR-1205 downregulation induces NEPC still remains elusive [47].

Enhancer of Zest Homologue 2 (EZH2) is an important factor involved in all key molecular processes in NEPC, including the loss of *TP53* and *RB1*, *NMYC* amplification, the activation of Wnt/β-catenin, and cell cycle arrest [15,30]. miRNAs involved in EZH2 axis have profound role in NEPC. Shan et al. reported that EZH2 binds to the promoter region of miR-708, inducing its silencing (Figure 1) [48]. The resulting downregulation of miR-708 induces NEPC in prostate adenocarcinoma. In prostate adenocarcinoma cells, miR-708 binds to sestrin-3 (SSN3) and inhibits FOXO1 phosphorylation, favoring apoptosis. However, in NEPC cells, cell cycle arrest and androgen deprivation inactivate CDK1 signaling while activating Wnt/β-catenin signaling. Together, this leads to the overexpression of EZH2 and the downregulation of miR-708. The sestrin-3-mediated phosphorylation of FOXO1 inhibits apoptosis and induces NEPC. As miRNA-708 is present in prostate adenocarcinoma cells but not in NEPC cells, its expression profile is a practical tool for predicting the stage and prognosis of NEPC [48].

Recent studies suggest that enzalutamide treatment enhances mast cell infiltration through the AR/IL-8 pathway [49]. When enzalutamide targets the AR, the cells evolve genetically to suppress AR expression and cue the recruitment of mast cells. The mast cell infiltration upregulates the expression of miR-32 and let-7a (Figure 1), leading to NE differentiation in PCa cells. Furthermore, miR-32 increases neuron-specific enolase expression, which suggests that miR-32 directly promotes NE differentiation, initiating NEPC [49]. In another study, Czyrnik et al. evaluated the role of miR-10b and miR-29a in NE trans-differentiated LNCaP cells and reported the induction of these miRNAs upon NED [50]. They further found that these miRNAs repress the hyaluronan synthetase (HAS3) gene [50].

### 3.2. MicroRNAs Regulating AR Signaling Pathway

Prostate cancer is an androgen-driven disease. Hence, the AR signaling pathway dictates tumor growth and cancer progression [51]. Novel molecular determinants involved in this signaling pathway provide insights into developing targeted therapy. FOXA1 is an important transcription factor that is involved in prostate epithelial differentiation, modulation of AR functions and AR signaling in prostate cells [52]. In PCa, FOXA1 directly inhibits the expression of IL8, a cytokine that is known to induce neuroendocrine differentiation. The loss of FOXA1 upregulates IL8, activating the MAPK/ERK pathway and ERK phosphorylation, inducing NEPC [53]. Recent studies found that miR-194 expression is inhibited via the transcription factor GATA2 in the AR signaling axis. miR-194 targets the FOXA1 gene. Altered FOXA1 gene expression subsequently derepresses important genes for cellular plasticity such as *SNAI2, ZEB1*, and *IL8* [54]. miR-204 also shows an AR-dependent expression pattern. AR downregulates the expression of miR-204 and thereby upregulates XRN1 and miR-34a (Figure 1). Enhanced miR-34a downregulates AR, forming a positive feedback loop for AR signaling in prostate adenocarcinoma and NEPC cells (Figure 1). Interestingly, miR-204 functions as a tumor suppressor in the former and as an oncomir in the latter tumor cell types [2]. In another study, miR-663 has been reported to be upregulated as LNCaP cells transition from androgen-dependent to androgen-independent state [55], with the concomitant upregulation of neuronal markers SYP, CHGA, and ENO2. miR-663 was shown to target dehydrogenase/reductase SDR-family member 7 (DHRS7), 24-dehydrocholesterol reductase (DHCR24), NKX3.1, and proteasomal subunit alpha type-7 (PSMA7) [55]. DHRS7 and DHCR24 may play a role in de novo androgen biosynthesis and metabolism in CRPC [55,56], whereas the altered expression of PSMA7 can increase AR transcriptional activity in CRPC [57]. 

### 3.3. MicroRNAs Regulating Other Genes Involved in NED

REST is a transcriptional repressor of NE-specific genes that is downregulated in NEPC. Liang et al. showed that REST is regulated by the miR-106b~25 cluster (miR-106b, miR-93 and miR-25) in PCa (Figure 1) [58]. Hypoxia treatment of neural crest (NC) cell cultures and the PCa cell line led to an upregulation of all the miRNAs of the miR-106b~25 cluster, suggesting that neuronal and neuroendocrine differentiation in neural crest cells and prostate cancer cells were promoted by hypoxia. REST also represses miRNA genes. In prostate tumors with a high Gleason score (≥8), an inverse correlation was observed between REST and miR-106b~25 induction. A subset of miRNAs including miR-9, miR-25, miR-30d and miR-302b were upregulated in high-Gleason score (≥8) PCa, suggesting a mechanism by which NED contributes to PCa malignancy. They proposed that the loss of REST and the induction of this set of microRNAs are potential novel clinical markers of advanced PCa [58].

Elevated levels of miR-21 in an androgen-dependent cell line was sufficient to drive its androgen-independent growth [59]. miR-21 is known to play oncogenic roles in PCa and its elevated levels in mouse models enhance tumor growth and lead to a castration-resistant phenotype [59]. Further, miR-21 confers resistance to docetaxel via direct targeting of PDCD4 (Figure 1) [60].

### 3.4. MicroRNAs Regulating Epithelial-to-Mesenchymal Transition, a Process That Drives NEPC

Cellular plasticity and the epithelial–mesenchymal transition (EMT) are essential for normal prostate development [36]. Essentially, the reactivation of the EMT confers cell motility, loss of cell adhesion, and restructuring of cytoskeleton, giving rise to an invasive tumor cell population that metastasizes to other sites. Further, the metastasized tumor cells lose mesenchymal cellular features by undergoing the reverse EMT process, i.e., the mesenchymal-to-epithelial transition (MET), which allows cellular adhesion and colony formation in the new site. Thus, the EMT facilitates tumor infiltration via molecular reprograming and cellular plasticity and aids in acquiring drug resistance [61].

Substantial data suggest that the activation of EMT accelerates PCa progression and leads to therapy resistance state [62,63,64]. Studies suggest that androgen deprivation therapy and/or inhibition of AR signaling promotes EMT in PCa [65]. Reportedly, a “master switch” that mediated the core EMT program was identified to induce NE differentiation in epithelial prostate cancer cells [63], linking the emergence of neuroendocrine differentiation states with the EMT.

The EMT is a complex cellular process orchestrated by a myriad of transcription factors and signaling pathways. Several transcription factors (TFs), such as the Snail family of transcriptional repressors SNAI1 (SNAIL) and SNAI2 (SLUG) and Zinc finger E box binding homeobox 1 and 2 (ZEB1 and ZEB2), are associated with EMT-mediated cellular plasticity. The pro-plasticity genes and their regulators are significant in neuro-trans-differentiation process and lineage switching [66,67]. As EMT confers migratory and invasive properties to tumor cells, the molecular signature of NEPC includes the upregulation of SNAIL1 (SNAIL), SNAIL2 (SLUG), ZEB1, and ZEB2 [66,67,68]. It has been reported that disrupting androgen signaling induces SNAI1 expression as AR directly represses *SNAI1* gene expression by binding to specific AR-responsive elements within the *SNAI1* promoter. As an adaptive response to enzalutamide, prostate cancer cells express SNAIL, which leads to EMT induction [67]. Another study showed that SNAIL overexpression in LNCaP cells results in EMT induction with expression of NED markers [66]. SNAIL inhibition in neuroendocrine-like PC3 cells abolishes attributes of NE differentiation and downregulates Sox2 [69].

Extensive studies have been performed to elucidate the role of miRNAs in regulating EMT. Prominent examples are the miR-200 family and miR-205, which regulate the EMT through the direct targeting of ZEB1 and ZEB2 (Figure 2) [70,71,72]. In androgen-independent cell lines, miR-200 is reported to suppress EMT by inhibiting ZEB1, ZEB2, and SLUG expression, resulting in decreased cell invasion and migration. miR-1 and 200b post-transcriptionally repress SLUG, while SLUG is a direct transcriptional repressor of miR-1 and 200b (Figure 2) [73], thus establishing a double-negative feedback loop between SLUG and miR-1, -200b [73]. A similar relation is observed between miR-145 and ZEB2 [74]. In addition, miR-205 has been shown to inhibit the EMT in PCa by directly targeting ZEB2 (Figure 2) among other targets, including N-chimaerin, ErbB3, E2F1, and E2F5 [75]. miR-205 overexpression in PCa cell lines reduced cell invasion by downregulating protein kinase Cε (PKCε) [75], a serine/threonine kinase member of the PKC subfamily that plays a role in PCa migration and invasion with its expression correlating with disease aggressiveness [76]. miR-301a activates the EMT by regulating the expression of β-catenin and E-cadherin (Figure 2) [77]. miR-301a overexpression is sufficient to induce neuronal genes in CRPC cell lines [26]. The loss of the miR-15a/miR-16-1 cluster, located on chromosome 13q14, has been reported to increase prostate cancer aggressiveness and play a synergistic role with upregulated miR-21 in promoting the EMT [78]. Das et al. elucidated the role of miR-194 in the prostate cancer EMT via its regulation of ubiquitin ligase suppressor of cytokine signaling 2 (SOCS2) (Figure 2) [79]. The ectopic delivery of miR-194 stimulated the EMT in human PCa cell lines and enhanced the metastasis of intravenous and intraprostatic tumor xenografts. SOCS2 downregulation by miR-194 resulted in derepression of the oncogenic kinases FLT3 and JAK2, leading to augmented ERK and STAT3 signaling. Further, they found miR-194 to be regulated by the transcription factor GATA2 [79].

Another molecular mechanism by which microRNAs regulate the EMT is via controlling alternative splicing in cancer cells. miR-375, together with the miR-200 family member miR-200c, directly regulates the expression of RNA-binding protein Quaking5 (QKI5) (Figure 2) [80]. QKI-5 directly binds to and regulates hundreds of alternative splicing targets. Thereby, it increases cell migration and invasion and restrains tumor growth. QKI-5 is necessary and sufficient to drive the alternative splicing events associated with EMT. Therefore, the miR-200/miR-375/QKI axis influences epithelial cell plasticity [80].

### 3.5. MicroRNAs Regulating Stemness Genes Involved in NEPC Trans-Differentiation

Prostate cancer cells with stem cell properties have long been implicated in tumor initiation and/or therapy resistance [36]. The induction of the EMT has been shown to generate cells with stem cell characteristics, including high metastatic competence and drug resistance [61]. Several studies support the view that stem cell signaling pathways fuel tumor plasticity in NEPC [36]. Androgen-dependent LNCaP cells and highly metastatic, androgen-independent PC3-M cells gain NE characteristics in response to various stimuli, including agents that augment intracellular cAMP (such as phosphodiesterase inhibitors), cytokines (such as IL6), or growth factors (such as EGF) [51,81,82]. The acquisition of forkhead transcription factor FOXC2 by AR+ prostate cancer cells led to NED and resistance to enzalutamide and docetaxel [63]. Activated Wnt signaling has been implicated in driving NE trans-differentiation in prostate cancer cells [83,84]. *NANOG, SOX2*, and *LIN28* are well characterized cell fate-determining pluripotent genes essential for pluripotency and the maintenance of self-renewal. SOX2 is a transcription factor involved in multiple signaling pathways, including the Wnt/β catenin, AKT/MTORC1, and TGFβ/SMAD pathways. It promotes lineage plasticity and anti-androgen resistance in PCa cells that lack *TP53* and *RB1* [85,86,87]. SOX2 plays a prominent role in NEPC progression because it binds to the β-catenin enhancer to activate the Wnt/β catenin pathway, promoting cell migration and invasion via the EMT. It is more highly expressed in NEPC, presumably due to the loss of *TP53* and *RB1* [88,89].

The signal transduction pathways involved in the EMT and stemness are common, such as TGFβ signaling, Wnt signaling, and Notch signaling [90]. Therefore, miRNAs with roles in the EMT have been linked to cancer stemness. As an example, miR-203 plays a role in prostate cancer EMT and stemness by targeting ZEB2 and Bmi1 [91]. It has been shown that miR-200c, miR-203, and miR-183 constitute stemness-inhibiting miRNAs that cooperate to suppress the expression of stem cell factors such as Bmi1 in cancer cells. Further, ZEB1 was found to link EMT activation and stemness maintenance by suppressing stemness-inhibiting miRNAs [68].

According to study by Lovnicki et al., a LIN28B/let-7/SOX2 axis contributes towards the progression of prostate adenocarcinomas to NEPC (Figure 2) [92,93]. They found that the LIN28B isoform and SOX2 upregulation were not mutually exclusive in NEPC cells. The co-upregulation was found to be mediated by let-7. The inhibition of let-7 promotes the derepression of HMGA2, resulting in SOX2 transcription and inducing cellular plasticity [92]. miR-320 is another stemness-regulating miRNA that is underexpressed in PCa [94]. It plays a tumor suppressor role in PCa, as its overexpression decreases PCa tumorigenesis in vitro and in vivo through targeting the LEF-1, CD44, SOX9, Oct-4, and CCND1, genes associated with the Wnt/β-catenin signaling pathway [94]. In addition, miR-320a has been reported to directly target lysosomal associated membrane protein 1 (LAMP1), which is overexpressed in CRPC clinical specimens [95].

A recent study showed that tumor stroma crosstalk mediated by miRNAs carried in exosomes play an important role in inducing PCa NED in response to hormone therapy [96]. The inhibition of AR signaling via enzalutamide treatment upregulates GRP78 (binding immunoglobulin protein), activates the Wnt/β-catenin pathway, and upregulates miR-29b in tumor cells. miR-29b is transported to stromal cells via extracellular vesicles (EVs), causing downregulation of the SPARC protein. Consequently, IL6, a known inducer of NED, is released [96].

## 4. MicroRNAs as Diagnostic Biomarkers for Neuroendocrine Prostate Cancer

Currently, the gold standard for NEPC diagnosis is pathology driven. Histopathological assessment combined with immunohistochemical detection in PCa tissues or serum levels of neuronal markers, including SYP, NSE, CHGA, and CD56, is used to monitor NED [97,98]. Neuronal markers commonly associated with NEPC, such as SYP, ENO2, and CD56, lack specificity [99,100]. Serum markers such as chromogranin A may be elevated and utilized for initial screening. Nonetheless, these markers are routinely high in patients taking proton pump inhibitors common in PCa therapy [101,102]. In NEPC, even the classical marker PSA is less accurate for diagnosis, staging, and monitoring cancer burden, as NEPC cells are AR independent. Herein, there is a need for the development of reliable biomarkers for the prediction, diagnosis, and prognosis of therapy-induced NEPC. Extensive studies are aiming to elucidate the signaling pathways that govern neuroendocrine differentiation and NEPC to develop adept diagnostics and treatments. The identification of reliable biomarkers is one of the primary goals of current research. Beltran et al. generated a 70-gene NEPC classifier that could distinguish CRPC-NE from CRPC-Adeno, treatment-naïve PCa, and BPH [13]. In another study, Aggarwal et al. developed a gene expression signature for t-SCNC [14]. As miRNAs are involved in important tumor-driving processes, such as cell proliferation, migration, invasion, and trans-differentiation, a detailed study of miRNA regulation and expression is needed for the development of biomarkers. miRNAs hold promise as multi-analyte markers that better reflect tumor complexity than mono-analyte markers [103,104,105].

Nanayakkara et al. performed miRNA profiling in neuroendocrine neoplasms (NENs) from different anatomical sites, including the gastrointestinal tract, pancreas, lung, parathyroid gland, pituitary gland, skin, thyroid gland, adrenal gland, and extra-adrenal sites. They reported that NENs can be distinguished from non-NENs based on miRNA profiles [105]. NENs from these diverse anatomical sites were found to have convergent miRNA expression programs, reflecting their morphological and functional similarities [105]. Comparative analyses showed that miR-375 and miR-7 expression is higher in NEN cases than non-NEN controls [105]. In prostate cancer, recent studies suggest the use of miRNA profiles as molecular tools to stratify CRPC and NEPC patient samples [26,106]. Small RNA next-generation sequencing (NGS) of clinical tissues from metastatic CRPC patients with adenocarcinoma features (CRPC-Adeno) vs. those with NE characteristics (CRPC-NE) showed that neuroendocrine trans-differentiation is accompanied by a characteristic set of miRNA alterations, including the downregulation of the miR-106a~363 cluster of miRNAs and the upregulation of miR-375 and miR-301a [26]. Unsupervised principal component analyses (PCA) of small RNA NGS data revealed separate clustering of the CRPC-NE tumors from CRPC-adenocarcinomas, suggesting that miRNA expression patterns can distinguish between CRPC-Adeno and CRPC-NE tumors. With application of machine learning algorithms to miRNA sequencing data, a ‘novel miRNA classifier’ was developed that could robustly stratify CRPC-NE tumors from CRPC-adenocarcinomas [26]. Notably, miR-375 was found to be a common miRNA between NEPC [26] and other NENs [105].

Ostano et al. combined miRNA profiles with the profile of protein-coding genes and lncRNAs to derive a ‘molecular classifier’ for NEPC [106]. Interestingly, the miRNA profiling in this study identified a set of six miRNAs as upregulated and one miRNA as downregulated in NEPC. Further, miRNAs alone could not stratify NEPC vs. adenocarcinomas [106]. This study utilized primary tumors (fresh frozen prostatectomy tissues) for their analyses as compared to FFPE tissues from metastatic sites employed in the study by Bhagirath et al. [26], suggesting divergent miRNA expression patterns between primary and metastatic tumors. In addition, the heterogeneity of the analyzed tumors may underlie these differences.

Exosomes are nano-sized vesicles typically between 30 and 150 nm in thickness that facilitate intercellular communication by carrying a cargo of proteins, microRNAs, mRNAs, or DNA from cells [107,108]. Exosome-mediated intercellular communication is pivotal for PCa progression and metastasis. Although exosomes were initially thought as ‘garbage bins’ to expel unnecessary and overproduced biomolecules out of the cell, they have opened new avenues for diagnostics and therapeutics [101,109]. Tumor cells secrete significantly more exosomes than normal cells [110]. Exosomes facilitate the tumor to reprogram the TME in favor of tumor growth. Reports suggest that exosomes are enriched in miRNAs and are a promising source of cancer biomarkers [111]. The analyses of exosomal miRNA expression profiles can provide valuable information regarding tumor cells and the TME. Examination of the miRNA content of exosomes in NEPC clinical samples showed a consistent dysregulation of key miRNAs and miRNA isoforms in NEPC as compared to corresponding adenocarcinoma cases [112]. Importantly, the application of machine learning algorithms identified a novel ‘EV-miRNA classifier’ that can be used to diagnose NED in CRPC patients non-invasively with important translational implications [112].

## 5. Interplay of MicroRNAs with LncRNAs

Long non-coding RNAs constitute an important class of regulatory RNAs that have been implicated in NEPC [57]. A set of 122 lncRNAs could robustly distinguish NEPC from prostate adenocarcinomas, demonstrating the potential clinical utility of lncRNA profiles [57]. Owing to the ability of lncRNA to form diverse complex tertiary structural complexes, lncRNAs can interact with miRNAs and transcription factors. The most studied interaction is between long non-coding RNAs and miRNAs, wherein lnRNAs can competitively bind to miRNAs and prevent their binding to cognate mRNAs, thereby relieving miRNA-mediated repression [113].

The lncRNAs HOTAIR and MALAT-1 were found to be upregulated in NEPC [106]. HOTAIR was reported to be repressed by repressor element-1 (RE-1) silencing transcription factor (REST) [114]. REST is downregulated in NEPC [115,116,117] that leads to the upregulation of HOTAIR, which is sufficient to induce PCa NED [114]. An interplay was observed between MALAT-1 and miR-1 in AR-independent PCa cell lines, where an inverse correlation between MALAT-1 and miR-1 was observed [118]. The authors propose that MALAT-1 acts as a molecular sponge of miR-1, resulting in the downregulation of KRAS in androgen-independent PCa cell lines [118]. In another study, lncRNA-p21 was reported to be highly expressed in NEPC PDX models. Its expression was enhanced by enzalutamide treatment, promoting the disruption of PRC2 complex and the EZH2-mediated methylation of STAT3 [119].

The lncRNA PCAT6 was found to be highly expressed in NE-like PCa cells and its levels were upregulated upon enzalutamide treatment [120]. PCAT6 knockdown in the NCI-H660 cell line repressed NED, while its upregulation in C4-2 cells promoted NED. Further, PCAT6 acted as a competing endogenous RNA (ceRNA) by sponging miR-326 and increasing the expression of its target protein Hnrnpa2b1 [120]. Mather et al. reported that LINC00261 was significantly upregulated in NEPC [121]. Knockdown of LINC00261 in PC3 cells attenuated its proliferative and metastatic abilities, which could be explained by the parallel downregulation of CBX2 and FOXA2 through distinct molecular mechanisms [121]. In the cell cytoplasm, LINC00261 binds to miR-8485 and titrates it away from binding to and repressing CBX2 mRNA. However, in the nucleus, LINC00261 could function as a transcriptional scaffold to induce SMAD-driven expression of the FOXA2 gene [121]. Zhu et al. showed that the lncRNA H19 and H19-derived miRNA, miR-675 was significantly downregulated in metastatic prostate cancer cell line as compared to a non-metastatic prostate epithelial cell line [122]. Furthermore, miR-675 directly binds to the 3′UTR of TGF-β1, an extracellular matrix protein, involved in cancer metastasis [122]. In conclusion, lncRNAs add an extra layer of regulation to the complexity of miRNA-mediated gene control.

## 6. Role of Circular RNAs in Neuroendocrine Prostate Cancer

Circular RNAs (circRNAs) are single-stranded, covalently closed RNA molecules that are highly stable and produced from mRNAs by backsplicing [123]. They play a role in gene regulation, acting as efficient miRNA sponges among other mechanisms [124]. Vo et al. created a landscape of circRNAs in prostate cancer [125]. They performed differential circRNA expression analyses in NEPC vs. CRPC samples and identified a set of 34 upregulated and 48 downregulated circRNAs in NEPC. In NEPC, circ-AURKA and circ-AMACR were the most significantly upregulated and downregulated circRNAs, respectively. These alterations in cirRNAs are consistent with alterations in the correpsonding genes, AURKA and AMACR, in NEPC [125]. Although circRNAs are efficient miRNA sponges [124], the correlation between circRNAs and miRNAs is unexplored in NEPC.

## 7. MicroRNAs as Therapeutic Targets for Neuroendocrine Prostate Cancer

Platinum-based drugs are currently employed clinically for the management of NEPC patients. Although targeted therapies such as DLL3 and AURKA inhibition are being tested as novel therapies for NEPC [126,127]; currently, there are no clinically implemented targeted therapies for NEPC. Given the important regulatory role of miRNAs in driving neuroendocrine phenotype in prostate cancer, miRNA-mediated signaling pathways can be potentially exploited for NEPC treatment. However, our understanding of miRNAs in NEPC is still in its infancy and would require a lot of focused work before miRNA-targeted therapeutics can be implemented. This would entail rigorous studies on miRNAs, especially in vivo studies. Further, the optimal routes of administration/delivery, doses and pharmacokinetics will have to defined [128]. While there are currently no clinical trials focusing on therapeutic application of miRNAs in NEPC, there are ongoing clinical trials involving miRNAs in diagnostics and disease monitoring in advanced prostate cancer. With increasing knowledge on the underlying mechanistic roles of miRNAs in NEPC, this understanding will hopefully be translated into defining effective targeted therapeutic interventions against the disease.

## 8. Conclusions and Future Perspectives

Increasing studies delineate the role of miRNAs as key epigenetic players in the emergence of lineage switching in prostate cancer. This will further our understanding of the complexities underlying NEPC states. These studies have unveiled an important role of cellular processes, such as EMT and cancer stemness, in determining cellular plasticity and neuroendocrine differentiation in PCa. In addition, these studies highlight the diverse roles that miRNA play as oncogenes and tumor suppressors in driving NEPC. However, we are far from a complete understanding of the complex role of miRNAs in PCa neuroendocrine differentiation. Studies so far have focused primarily on in vitro approaches. To comprehend the physiological roles of miRNAs in PCa trans-differentiation, in vivo studies are needed. Furthermore, a systems biology approach may yield useful insights into the concerted role of miRNAs in NEPC. A detailed mechanistic understanding will contribute to better diagnosis and targeted therapy for this aggressive variant of prostate cancer. These studies will pave the way to translate the applications of miRNA into the clinical setting.

## Figures and Tables

**Figure 1 ncrna-08-00025-f001:**
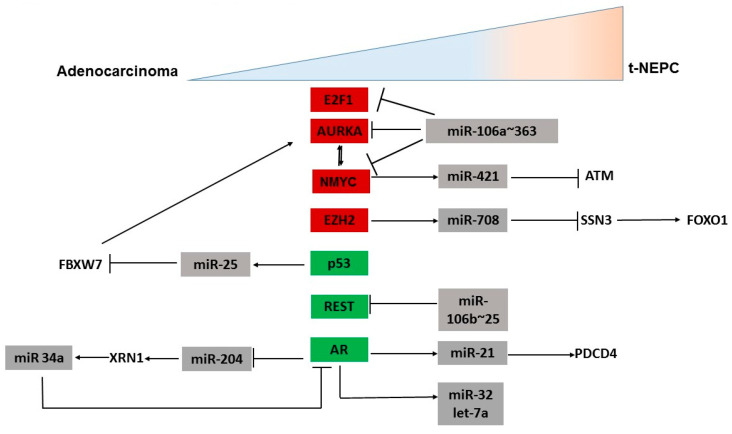
MicroRNAs impact key genes implicated in neuronal differentiation of prostate cancer cells. Schematic representation of miRNA–mRNA interactions underlying neuroendocrine differentiation in prostate cancer cells. Upregulated key genes in NEPC are highlighted in red and downregulated key genes are highlighted in green.

**Figure 2 ncrna-08-00025-f002:**
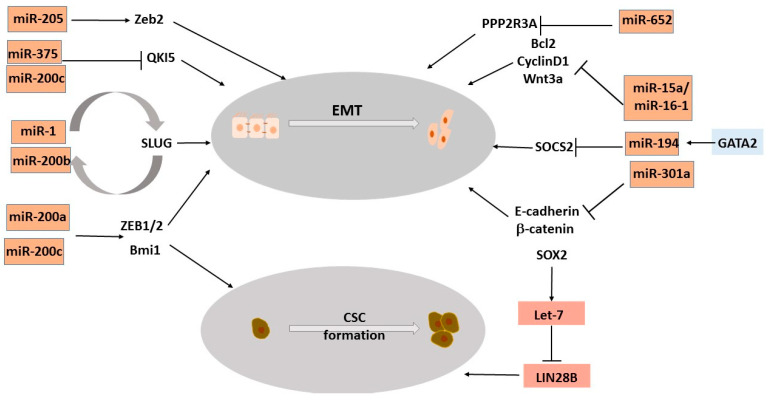
MicroRNAs impact EMT and cancer stemness. Schematic representation showing miRNAs affecting mRNAs of genes implicated in epithelial-to-mesenchymal transition and cancer stem cell formation.

## Data Availability

Not applicable.

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
