# Peer review of "Role of MicroRNAs in Neuroendocrine Prostate Cancer"

_ncrna, 2022, doi:10.3390/ncrna8020025_

Round 1

Reviewer 1 Report

Authors enhanced tremendously the value of this manuscript. Many additional findings were included and not only listed but also put togteher and discussed.

Author Response

We sincerely thank the reviewers and editorial board for their constructive comments. We have thoroughly checked and revised the manuscript as suggested to include language corrections. The revised file has tracked changes. We sincerely hope that it is acceptable for publication.
We again thank the reviewers for all the comments.

Reviewer 2 Report

The authors did a good job at responding to my comments

Author Response

We sincerely thank the reviewers and editorial board for their constructive comments. We have thoroughly checked and revised the manuscript as suggested to include language corrections. The revised file has tracked changes. We sincerely hope that it is acceptable for publication.
We again thank the reviewers for all the comments.

This manuscript is a resubmission of an earlier submission. The following is a list of the peer review reports and author responses from that submission.

Round 1

Reviewer 1 Report

This review is reasonably coherent and would be of some interest to the field. It could be readily improved by addressing the following points:

  1. Section 3 needs to describe the evidence for miRNAs influencing NED by targeting genes involved in neuroendocrine plasticity. The problem with the current structure is that the authors describe how miRNAs influence EMT and stem-ness before discussing how they regulate NED. In other words, much of the text in Figure 3.3/3.4/3.5 should come prior to 3.1/3.2.
  2. The authors are correct to include sections describing the roles of miRNAs in EMT and stem-ness, since EMT and stem-ness are other facets of lineage plasticity that are closely related to neuroendocrine plasticity. However, since this review is on the role of microRNAs in NEPC, the authors must have at least a brief preamble describing the links between neuroendocrine, mesenchymal and stem-like phenotypes.
  3. Key references in the “EMT” section are missing. Multiple other miRNAs (e.g. miR-194, miR-375) have been demonstrated to regulated EMT in prostate cancer.
  4. Much of section 4 belongs in section 5, since it is describing the potential of diagnosing NEPC using miRNAs (albeit in exosomes). These 2 sections should be combined and ref 33 incorporated into an earlier, more appropriate section of the manuscript. This would also overcome the issue of having an entire section devoted to exosomal miRNAs in NEPC, which is unnecessary.
  5. The authors tend to describe their own studies in more detail than other seminal papers in the field, which is inappropriate. This is particularly evident in sections 4 and 5.
  6. Figure 1: This figure would be a lot more compelling if it showd how miRNAs target specific mRNAs to influence prostate cancer cell plasticity and NED. Also, it would be useful if this figure were labelled “Figure 1” and better linked to the text/legend.
  7. A section describing the potential of targeting miRNAs as a therapeutic strategy for NEPC is warranted. Although it is true that miRNA-based therapies have not been successfully transitioned through clinical trials, we must be hopeful and forward-thinking!
  8. Important statements lack citations e.g. “t-NEPC is also seen in patients after primary docetaxel therapy”.

Reviewer 2 Report

The review by Sreekumar and Saini summarizes the role of miRNAs in a subtype of prostate cancer with neuroendocrine phenotype and suggest a reversible trans-differentiation process. Authors list potential oncomiRs and microRNAs that mediate the crosstalk between tumor cells and the tumor microenvironment as well as associations of miRNA expression to neurendocrine phenotype. However, neither a direct role of miRNA to this phenotype is known nor being regulating a reversible trans-differentiation. Thus, the review is more speculative and may mislead readers.

Aspects are missing such as interaction of miRNA with lncRNAs and circRNAs. Further, relevant study findings, identifying the connection/discrepancies among studies and put in the reviewer’s own perspective must be persued, instead of simply listing out previous findings as this manuscript did for most of the time.

Several recent contributions have summarized in a more comprehensive manner this topic:

DOI: 10.1016/j.gene.2020.144995, doi: 10.1038/s41388-020-01493-8, doi: 10.20517/cdr.2020.30

Thus, there iare no significant novelties in the submitted manuscript.